# Follow-up of a historic cohort of children treated for severe acute malnutrition between 1988 and 2007 in Eastern Democratic Republic of Congo

**Pacifique Mwene-Batu**[1,2,3,4]*, **Ghislain Bisimwa**[1,3], **Gaylord Ngaboyeka**[1,4], **Michelle Dramaix**[2], **Jean Macq**[5], **Daniel Lemogoum**[6], **Philippe Donnen**[2]

**1** Ecole Régionale de Santé Publique, Université Catholique de Bukavu, Bukavu, Democratic Republic of Congo, **2** Ecole de Santé Publique, Université Libre de Bruxelles, Brussels, Belgium, **3** Nutritional department, Centre de Recherche en Sciences Naturelles, Lwiro, Kinshasa, Democratic Republic of Congo, **4** Hôpital Provincial General de Reference de Bukavu, Université Catholique de Bukavu, Bukavu, Democratic Republic of Congo, **5** Institute of Health and Society, Université Catholique de Louvain, Brussels, Belgium, **6** Hôpital ULB-Erasme, Université Libre de Bruxelles, Brussels, Belgium

* lyabpacifique@yahoo.fr

**Data Availability Statement:** All relevant data are within the paper.

## Abstract

### Background

It is well documented that treatment for severe acute malnutrition (SAM) is effective. However, little is known about the long-term outcomes for children treated for SAM. We sought to trace former SAM patients 11 to 30 years after their discharge from hospital, and to describe their longer-term survival and their growth to adulthood.

### Methods

A total of 1,981 records of subjects admitted for SAM between 1988 and 2007 were taken from the archives of Lwiro hospital, in South Kivu, DRC. The median age on admission was 41 months. Between December 2017 and June 2018, we set about identifying these subjects (cases) in the health zones of Miti-Murhesa and Katana. For deceased subjects, the cause and year of death were collected. A Cox proportional hazards multivariate regression analysis was used to identify the death-related factors. For the cases seen, age- and gender-matched community controls were selected for a comparison of anthropometric indicators.

### Results

A total of 600 subjects were traced, and 201 subjects were deceased. Of the deceased subjects, 65·6% were under 10 years old at the time of their death. Of the deaths, 59·2% occurred within 5 years of discharge from hospital. The main causes of death were malaria (14·9%), kwashiorkor (13·9%), respiratory infections (10·4%), and diarrhoeal diseases (8·9%). The risk of death was higher in subjects with SAM, MAM combined with CM, and in male subjects, with HRs* of 1·83 (p = 0·043), 2·35 (p = 0·030) and 1·44 (p = 0·013) respectively. Compared with their controls, the cases had a low weight (-1·7 kg, p = 0·001), short

**Funding:** This study is part of a Research for the Development Project entitled: 'Implementation study of a psycho-medico-social care model at the health centre level: the case of people with chronic diseases and the mother-child malnourished couple, South Kivu, Democratic Republic of Congo' and funded by Belgian Development Cooperation through the Académie de Recherche et d'Enseignement Supérieur (ARES). The funder of the study had no role in study design, data collection, data analysis, data interpretation, or writing of the report.

**Competing interests:** The authors have declared that no competing interests exist.

height [sitting (-1·3 cm, p = 0·006) and standing (-1·7 cm, p = 0·003)], short legs (-1·6 cm, p = 0·002), and a small mid-upper arm circumference (-3·2mm, p = 0·051). There was no difference in terms of BMI, thoracic length, or head and thoracic circumference between the two groups.

## Conclusion

SAM during childhood has lasting negative effects on growth to adulthood. In addition, these adults have characteristics that may place them at risk of chronic non-communicable diseases later in life.

## Background

Malnutrition in all its forms is a global public health problem. In low- and medium-income countries (LMIC), excess weight and obesity among adults is rapidly rising. Their prevalence exceeds that of undernourishment. However, undernourishment still largely prevails among children in these regions [1–4].

In 2015, the Millennium Development Goals (MDG) report showed that the proportion of children who are underweight worldwide fell overall from 25% in 1990 to 15% in 2015 [5,6]. However, this decrease was not equal across all the world's regions [7]. In sub-Saharan Africa, only minimal progress had been made. Whereas Southeast Asia, despite the biggest decrease in the proportion of underweight children since 1990, had the highest underweight prevalence in 2015 [6,7].

Globally, sub-Saharan Africa had the lowest decrease in the hunger index between 1990 and 2012 [6]. However, in 2018, the Food and Agriculture Organization of the United Nations (FAO) showed that the number of undernourished people in the world had increased in the last three years, returning to the level recorded almost ten years ago. Furthermore, the downward trend in the undernourishment rate in Asia seems to be slowing considerably [8].

Considerable progress has been made in the treatment of acute malnutrition (AM). Starting with a purely medical model, a proven approach referred to as Community Management of Acute Malnutrition (CMAM) was introduced in 2001 to decentralise therapeutic care to the community [9–11]. This approach has been implemented in emergency and routine situations in various countries and there is now a great deal of data demonstrating the success of this model [9,10,12,13].

Studies have also shown that subjects who suffered severe acute malnutrition were always exposed to a high risk of relapse or mortality in the short term even after nutritional rehabilitation because in the majority of cases these subjects would return to the same precarious living conditions [14–19].

Despite the successful short-term outcome of severe acute malnutrition (SAM) treatment, its long-term assessment is difficult to measure. To date, few studies have focused on the long-term outcomes for former SAM subjects after their nutritional rehabilitation in LMIC. The few studies that have examined this issue are mostly old and concerned subjects who were prepubescent at the time of follow-up [9,17,18]. This is a major gap for science.

Moreover, these subjects are at major risk of adult-onset non-communicable diseases (NCD) as suggested by the Barker hypothesis [20,21]. According to this hypothesis, intra-uterine foetal adaptations occur in response to foetal malnutrition induced by limited protein in the mother during pregnancy (observed in rats) leading to metabolic and structural changes that have short-term benefits but increase the risk of adult-onset NCDs [20].

Despite the evidence in animals, this is harder to demonstrate in humans given the unethical aspect of this kind of intervention study. Subsequently, periods of famine such as those in the Netherlands during the Second World War (Dutch famine), and in China between 1959 and 1960, offered the unique opportunity to investigate this link in high- and medium-income countries (HMIC) [19,21–25].

In HMIC, it has been shown that subjects with a low-birth weight, low weight gain in early childhood, followed by rapid weight gain during puberty were more exposed to metabolic syndrome [26–27], diabetes [28,29], and to high levels of visceral fat in adulthood [29]. In addition, they were susceptible to being smaller and less productive, having low educational attainment, a low socio-economic status in adulthood, and giving birth to underweight children [30,31].

Despite growing evidence on the negative long-term effects of undernourishment during childhood in HMIC, little is known about the long-term outcomes for children treated for SAM in low-income countries (LIC) [9,18].

This is because recreating cohorts of adults who suffered from SAM in childhood is a major challenge in LIC, given that archiving is a problem, as are the frequent population movements. In addition, extrapolating the results for HMIC to LIC is questionable because HMIC differ from LIC in many ways (lifestyle, cause of low-birth weights, eating habits, etc.). The resulting problem, therefore, is lack of data on the potential long-term health and economic productivity consequences of childhood SAM in LIC.

In the Democratic Republic of the Congo (DRC), malnutrition is still a major public health problem, with a chronic malnutrition prevalence of 43% among children under the age of five [32]. In 2017, UNICEF (United Nations International Child Emergency Fund) estimated the number of children with SAM at 1.9 million and with moderate acute malnutrition (MAM) at 1.5 million [5].

In South Kivu, one of the 26 provinces of the DRC (located in the east of the country), malnutrition has been endemic since the 1960s [33]. One in two children under the age of 5 has chronic malnutrition (CM) and 7.9% have acute malnutrition (AM) [5,32]. The persistence of armed conflict over the past twenty years, limited accessibility of quality healthcare for the majority of the population, difficult access to farmland and inadequate nutrition (monotonous, undiversified and poor quality) are the main cause of this [34–36].

Lwiro paediatric hospital (HPL) was one of the first facilities to be involved in treating malnutrition in the DRC. A team of researchers supported by the Centre Scientifique et Médical de l'Université Libre de Bruxelles pour ses Activités de Coopération (CEMUBAC) developed a SAM treatment model in the 1980s, and began digitising clinical data in 1986. The electronic records contain sociodemographic, anthropometric, clinical and biological data gathered from inpatients between 1988 and 2007, from admission through to discharge from hospital.

Even though the treatment programme led to the recovery (nutritional rehabilitation) of most of the children, their medium- and long-term nutritional and health outcomes remain unknown.

Our objective, therefore, was to trace these patients, who were treated for SAM at Lwiro hospital, 11 to 30 years after their nutritional rehabilitation, to describe their longer-term survival and their growth to adulthood.

## Methods

### Study design and population

Our study population was made up of subjects admitted for SAM between 1988 and 2007 to Lwiro hospital (HPL) in the province of South Kivu in the DRC. The study subjects were identified using the HPL's database and sought in their villages of origin. They were then divided

into four categories (living in the village or the surrounding area, deceased, moved, or lost to follow-up). For each case seen, a community control was randomly selected to compare growth. The control was defined as a subject who had no history of SAM, was the same gender, was living in the same community, and was no more than 24 months older or younger than the case subject. To identify these control subjects, Community Health Workers (CHWs) spun a bottle at the case subject's home, then went door to door starting with the nearest house in the direction shown by the bottle until they found a subject that met the criteria. Initially, we wanted a control for each case. However, the control subjects were harder to recruit than the case subjects because many feared being associated with childhood malnutrition because of its social stigma. Unfortunately, their selection was limited to the number of eligible adults in the community. That is why we only obtained controls for three quarters of the cases.

At that time, diagnosis of SAM at the HPL was based on the weight-for-height ratio plotted on the local child growth curve established by DeMaeyer in 1959 and unpublished [37], the presence of nutritional oedema, and on serum albumin levels (by zone electrophoresis).

According to these criteria, a distinction was made between the following forms of malnutrition [38,39]:

1. Kwashiorkor: weight-for-height > 5th percentile, presence of nutritional oedema, and/or serum albumin < 30 g/l

2. Marasmic kwashiorkor: weight-for-height < 5th percentile, presence of nutritional oedema, and/or serum albumin < 30 g/l

3. Marasmus: weight-for-height < 5th percentile, absence of nutritional oedema, and serum albumin > 30 g/l

Nutritional therapy has changed over the years, with three distinct periods. During the first period (1987–1994), treatment was based on MASOSO gruel, which is a blend of corn, soy and sorghum. A key feature in the second period (1994–1996) was the administration of locally produced high-energy milk (HEM), which was a mixture of milk, oil and sugar and had an energy density close to 90 kcal/liter. During the third period (August 1996–December 2007), HEM was replaced by the therapeutic milk F-75 (in the 1st phase of treatment) and F-100 (in the 2nd phase) [39].

### Study framework

The study was conducted at the Centre de Recherche en Science Naturelle de Lwiro (CRSN-Lwiro), in the health zones (HZ) of Katana and Miti-Murhesa in South Kivu. The HZs of Miti-Murhesa and Katana are located 33 and 40 km from the city of Bukavu (capital of the province of South Kivu) respectively.

The CRSN was created in 1947 and its activities are grouped under four research departments: biology, geophysics, nutrition and documentation. The Nutrition Department has a paediatric hospital and several integrated health centers which monitor the state of health and nutrition of children in the community.

The HPL, which is a 70-bed hospital, is located 50 km from the city of Bukavu. In the 1970s, it operated as a nutritional rehabilitation center that only admitted children who had SAM. Since the 1980s however, it has been considered as a referral hospital for all paediatric conditions. Research activities were also carried out at the hospital. It had between five and seven doctors. Consultations were given by a general practitioner under the supervision of a paediatrician [39].

The main economic activities of the population in the HZs of Miti-Murhesa and Katana are agriculture, livestock farming, fishing and small trade.

Agriculture and livestock farming, once considered as primary sources of income for nearly 70% of the population, have been heavily disrupted by the insecurity and its consequences (pillaging and displacements) [33]. The staple food in the zone is cassava fufu with cassava leaves. In the zone, the population generally eats twice a day, except during the lean season, between July and August, when the number of meals drops to one a day [33]. Livestock farming concerns cows, goats and pigs.

Fishing is traditional-style fishing on Lake Kivu, but without equipment appropriate for its development [33]. Lastly, small trade is confronted with the challenge of 'transit taxes', imposed at many roadblocks on the main roads in the zone, leading to an increase in the price of foodstuffs [33].

## Outcomes

Our outcomes of interest were the longer-term survival (and its corollary mortality and its causes) and the long-term growth of subjects who had suffered SAM during childhood.

Survival was defined as a subject who was traced and seen or whose relatives reliably reported that he or she was alive. For mortality, the parameters considered were cause of death and age on death. For long-term growth, we considered anthropometrics in adulthood.

For the causes of death, as the time that had elapsed between the majority of deaths and the study made it impossible to use the WHO (World Health Organization) verbal autopsy due to significant memory bias, we considered the probable causes mentioned by close relatives. We then compared these with information gathered from the death registers and from the medical records available in the zone's health facilities. Age on death was deduced from the year of death provided by the nearest relative and the date of birth given in the HPL's records.

Long-term growth was assessed by anthropometrics in adulthood both for the case subjects and for the control subjects.

The parameters used were: weight (kg), height when sitting and standing (cm), leg and thoracic length (cm), head and thoracic circumference (cm) and the mid-upper arm circumference (MUAC) in mm. The BMI (Body Mass Index) was calculated using the formula weight/height$^2$ in kg/m$^2$ and was grouped into four categories: < 18·5 = underweight, 18·5 to 24·9 = normal, 25 to 29·9 = overweight, and ≥ 30 = obese.

The nutritional status of the study subjects at the time of their hospitalisation was reassessed in relation to the WHO child growth standard of 2006 [40]. A new classification was established according to the following criteria:

A child was classed as having SAM if they had met at least one of the three following criteria: a MUAC of < 115 mm, a weight-for-height of < -3 Z-score and the presence of nutritional oedema in the hands and/or feet and/or face. Kwashiorkor was defined by the presence of nutritional oedema in the hands and/or feet and/or face. Marasmus was based on a MUAC of < 115 mm and/or a weight-for-height of < -3 Z-score without nutritional oedema, and the mixed form was defined by the presence of nutritional oedema with either a MUAC of < 115 mm or a weight-for-height of < -3 Z-score or both. MAM was defined by a weight-for-height of between -3 and -2 Z-score and/or a MUAC of between 115 and 125 mm and without oedema. Lastly, CM was defined by a height for age of <-2 Z-score [40]. This new classification of nutritional status was the only one used for the rest of the analysis.

## Data collection

After a detailed analysis of the Nutrition Department's archives for the period 1988 to 2007, a total of 2,830 medical records of children admitted for SAM according to the criteria at that time were found and examined. Each record contained information on the subject's identity

(name of the child and their parents, date of birth, gender, health district of origin, and ethnicity), their vaccination schedule, anthropometric parameters (weight, height and MUAC), serum albumin levels, the physical examination on admission, the nutritional diagnosis, the treatment provided and the discharge from hospital.

The criteria for inclusion in the study were that participants had a record containing the above information, came from a health district in Miti-Murhesa or Katana, and were at least 16 years old at the time of recruitment in adulthood. The age of 16 was selected because, at this age, most individuals are already pubescent.

The following were excluded from our study: records of patients who left or died during hospitalisation, those of patients transferred to another facility, and records of adults admitted for SAM.

Based on these criteria, out of the 2,830 records, only 1,981 records (70%) were selected for the study.

Over the course of 12 months (December 2017 to November 2018), 20 CHWs and two supervisors worked full-time, with the participation of village leaders, to trace these subjects who were now adults and still living in Miti-Murhesa and Katana. To identify them, the CHWs used their identity, that of their parents, and the health district of origin noted in the hospital record. Once in the health district, they got in touch with neighbourhood leaders, registered nurses and community liaisons and gave them the identity of the subject and their parents. These informants would then give the CHWs the likely address of the parents. The CHWs were accompanied by a guide, selected from among the influential people of the village, to facilitate their contact with the households and the cooperation of the families. The guide gave prior warning to the target family members in the various villages about the CHW's visit in order to ensure that they would be present and informed about the team's intentions.

When the CHWs arrived at the family home, they spoke with a member of the family or the nearest neighbour they could find in order to identify the subject in question. If the subject was no longer living in the family home, the CHWs had to search for them using the new address provided. The CHWs often took two to three days to find a single subject and had to walk more than 20 km on foot in the hope of finally finding the subject. As this is a rural region where agriculture plays a major role, the CHWs had to start the identification very early, at about 5 a.m., to finish by 9 a.m., then continue their work at about 5 p.m. when most of the subjects were coming back from the fields to finish at 7 p.m. given the insecurity and lack of street lighting. If the subject had left the study zone, the CHWs noted the new zone of residence. Unfortunately, due to lack of financial means, we could not go outside the two study zones, which are relatively easy to access and relatively stable as regards security.

For the subjects that were seen, the CHWs took anthropometric measurements both for them and for their community controls.

The anthropometric measurements were taken according to WHO guidelines [40] and were quality controlled, meaning that they were taken independently by two members of the team. The final measurement was the average of the two. In the event of a difference greater than 0·5 cm or 300 gr, a third measurement was taken and the final measurement was the average of the two closest measurements.

For deceased subjects, the CHWs spoke with the closest relative who had lived with the deceased during the last moments of their life in order to accurately identify the year of death, the circumstances surrounding the death, the health facility in which the death occurred, and the probable cause of death given by the doctors for deaths that occurred in hospital settings. This information was compared with that provided in the death registers, and medical records if the deceased had been in contact with a health facility before their death. In the event of disagreement (a few cases), we took the diagnosis given by the health facility.

However, for 34 deaths, no information was available due to lack of death registers in the health facilities (24 deaths), or because the deaths occurred at home abruptly with no apparent reason (7 deaths) or a probable poisoning according to the relatives (3 deaths).

## Statistical analysis

Length/height-for-age, weight-for-length/height, body-mass index-for-age and weight-for-age Z-scores were calculated by using the WHO Child Growth Standards and ENA for SMART software, version October 2007 [40,41]. The analysis was performed by using the SPSS version 25.0. Sample size was predetermined by the number of medical records for children admitted for SAM from 1988 to 2007. Categorical variables were summarised as frequency and proportion and quantitative data as mean and standard deviation (SD) or median and min-max depending on whether the distribution was symmetrical or not. We used the Pearson Chi$^2$ test to compare the proportions for the categorical variables. For the long-term growth analysis, we compared all cases to community controls using the student t test analysis. For survival analysis, time 0 was the date of discharge to Lwiro Hospital. Time to follow-up for patients was calculated in years by subtracting date of discharge from date of final outcome (alive or dead). The Cox regression was used to identify factors influencing survival. In multivariate analysis, we adjusted for age at admission to hospital, nutritional status, vaccination schedule and sex. The proportionality hypothesis of instantaneous death risks (Proportional Hazards = PH model) was verified by the ln (-ln (S (t)) curves for each variable. Adjusted HR (Hazard Ratio) is followed by the p value of the Wald test. Since no deaths occurred among children without malnutrition, we combined this group with that of stunted children to form a new group of no acute malnutrition subjects that served as the reference group for the nutritional status. In all analyses, we deemed a p value < 0·05 as showing a statistically significant difference between groups.

## Ethical considerations

Respondents provided singed informed consent for participation in the study, either by written signature or by fingerprints, depending on literacy. For Child below 18 years of age, consent was obtained from the children's parent or guardian. Ethical approval for the study was obtained from the Université Catholique de Bukavu ethics committee.

## Results

### Demographic and health characteristics of the case group on admission at HPL

Out of the 2,830 medical records found in the HPL's archives for 1988 to 2007 and reviewed, only 1,981 (70%) met the criteria for inclusion in this study (Fig 1). On admission to hospital, the median age was 41 months, with 70·8% of patients aged between 6 and 59 months old. There were more boys (57·5%). Nearly three quarters of the patients were not up to date with their vaccinations (Table 1).

Table 2 presents the classification of the nutritional state of the children selected (cases) based on the criteria used at the time of hospitalisation in Lwiro [37] and on the current WHO criteria [40] respectively. Based on the WHO child growth standard, only 84% of the children were classed as having SAM. The others were classed as having MAM (6·7%) and not suffering from AM (9·3%). Nearly 90% of the children admitted for SAM at the HPL also had CM.

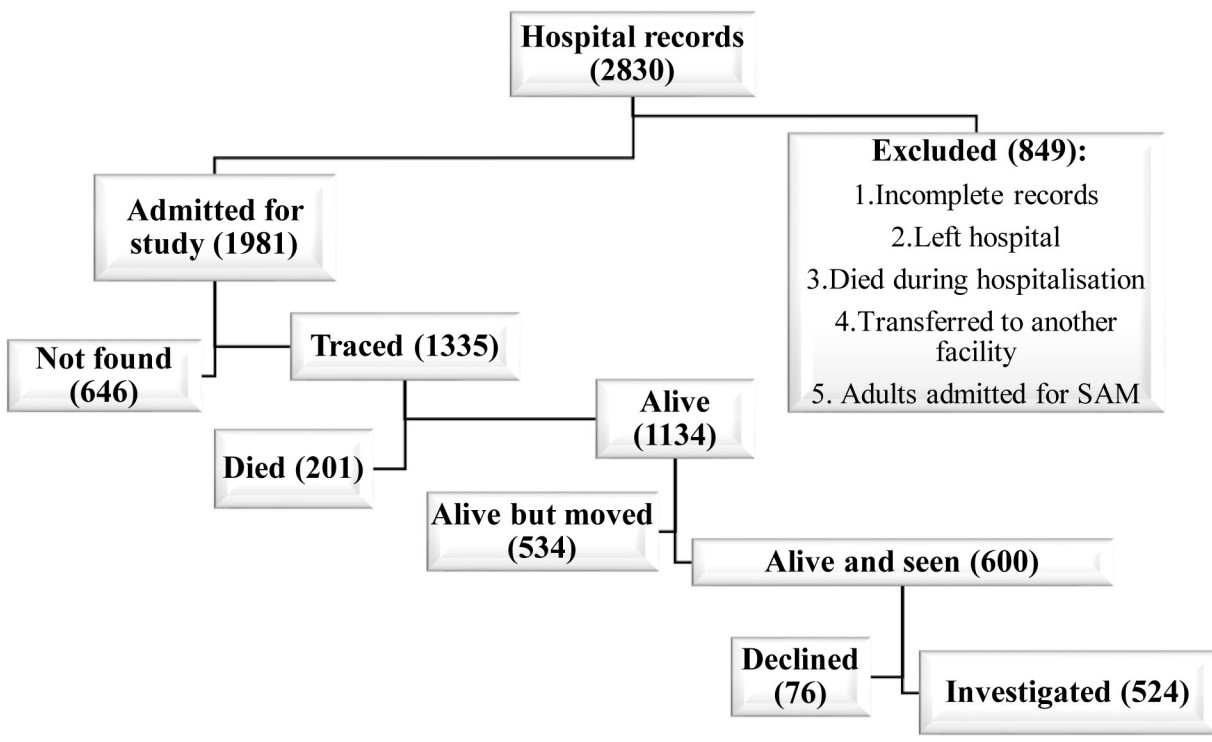

**Fig 1. Recruitment of the case group.**

### Recruitment of the case group

Out of the 1,981 subjects hospitalised, 1,335 (67·4%) were traced and 646 (32·6%) were lost to follow-up. Among those traced, 1,134 subjects (84·9%) were still alive and 201 (15·1%) were deceased. Among the living, 600 (52·9%) were seen by the CHWs and 534 (47·1%) had moved to other regions. Out of the 600 subjects seen, 524 agreed to participate in the study and 76 declined (Fig 1).

By comparing the hospital data of those lost to follow-up (n = 646) and those traced (n = 1335), we saw no difference in terms of nutritional state, age at which SAM occurred, gender, or vaccination status (Table 3).

**Table 1. Demographic and health characteristics of the case group on admission to the hospital.**

|  | % |  | Med (Min-Max) |
|---|---|---|---|
| **Number of children (N)** |  | 1981 |  |
| **Children age (Months)** |  |  | 41 (0–144) |
| < 6 | 1·9 |  |  |
| 6–59 | 70·8 |  |  |
| > 59 | 27·3 |  |  |
| **Gender** |  |  |  |
| Boy | 57·5 |  |  |
| Girl | 42·5 |  |  |
| **Vaccination Schedule** |  |  |  |
| Up to date | 28·0 |  |  |
| Not up to date | 72·0 |  |  |

**Table 2. Classification of nutritional status of the selected children (cases) based on the classification criteria used at the time of hospitalisation and on the 2006 WHO child growth standard respectively.**

|  | Criteria used at HPL[*] | WHO child growth standard |  |
|---|---|---|---|
| **Number of children** | % | % | N = 1981 |
| **Acute malnutrition** |  |  |  |
| No Acute Malnutrition |  | 9·3 |  |
| MAM˚ |  | 6·7 |  |
| SAM˚ | **100** | 84 |  |
| Marasmus | 21·4 | 9·7 |  |
| Kwashiorkor | 70·8 | 70·8 |  |
| Mixed | 7·8 | 3·6 |  |
| **Stunting** |  |  |  |
| No stunting |  | 10·6 |  |
| Stunting |  | 89·4 |  |

˚MAM = Moderate Acute Malnutrition, ˚ SAM = Severe Acute Malnutrition

[*]HPL = Hôpital Pédiatrique de Lwiro (Lwiro Paediatric Hospital)

## Long term post-treatment survival and causes of death

A third of the deaths occurred in the two years following discharge from hospital and more than half within five years (Fig 2). Fig 3 shows that approximately two thirds of the deceased subjects were aged ten or under. The main causes of death were malaria, kwashiorkor, respiratory infections and diarrhoeal diseases (Table 4).

## Main predictor of death

The proportion of deaths since discharge from hospital was higher in children who at the time of admission had MAM with or without CM, and lower in children who did not have AM. For SAM with or without CM, mortality was intermediate.

**Table 3. Comparison of hospital data of the subjects traced and those lost to follow-up.**

|  | Traced (N = 1335) | Lost to follow-up (N = 646) | P value |
|---|---|---|---|
|  | % | % |  |
| **Nutritional status (based on WHO standard)** |  |  |  |
| No Malnutrition | 1·1 | 2·2 |  |
| MAM only | 1·2 | 0·8 |  |
| SAM only | 7·8 | 8·7 | 0·40 |
| Stunting only | 9·0 | 8·8 |  |
| Stunting and MAM | 5·6 | 5·4 |  |
| Stunting and SAM | 75·3 | 74·1 |  |
| **Gender** |  |  |  |
| Male | 57·0 | 58·5 | 0·52 |
| Female | 43·0 | 41·5 |  |
| **Vaccination schedule** |  |  |  |
| Up to date | 26·7 | 30·5 | 0·08 |
| Not up to date | 73·3 | 69·5 |  |
| **Age range (Months)** |  |  |  |
| < 6 | 2·3 | 1·2 | 0·26 |
| 6–59 | 70·3 | 71·7 |  |
| > 59 | 27·3 | 27·1 |  |

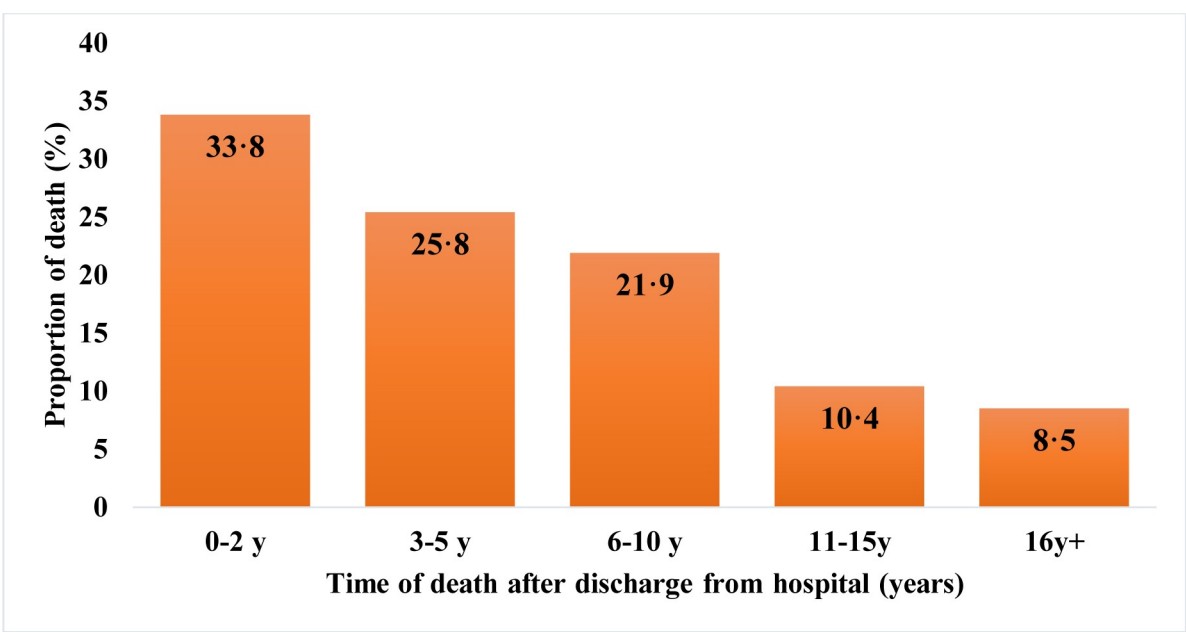

**Fig 2. Proportion of deaths according to time since discharge from hospital (n = 201).**

Mortality among the subjects who were under 6 months at the time of their hospitalisation was half as low as that of older children. Mortality in men was higher than in women and it was higher in subjects with a vaccination schedule that was not up to date compared with those who had an up-to-date vaccination schedule (Table 5).

In the Cox multivariate analysis (Table 5), the risk of death was statistically higher in subjects with SAM or MAM associated with CM and in male subjects. The immune status and age at which AM occurred did not have any significant influence on the risk of death.

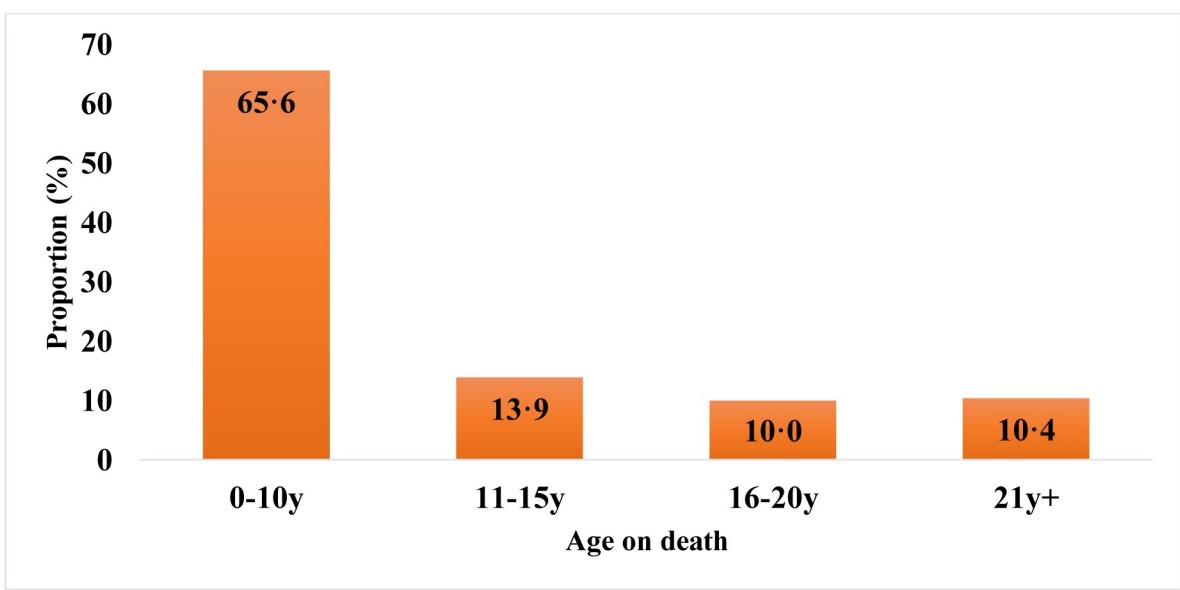

**Fig 3. Age on death after discharge from hospital (n = 201).**

**Table 4. Cause of death after discharge from hospital.**

| Causes of death | N | % |
|---|---|---|
| Malaria | 30 | 14·9 |
| Kwashiorkor | 28 | 13·9 |
| Respiratory infection | 21 | 10·4 |
| Diarrhoeal diseases | 18 | 8·9 |
| Anaemia | 13 | 6·5 |
| Meningitis | 11 | 5·5 |
| Road traffic accident | 10 | 5·0 |
| Diabetes | 9 | 4·5 |
| HIV/AIDS* | 3 | 1·5 |
| Others causes | 24 | 11·9 |
| Unknown causes | 34 | 16·9 |
| Total | 201 | |

*HIV: Human Immunodeficiency Virus, **AIDS**: Acquired Immune Deficiency Syndrome

## Long term growth after nutritional rehabilitation

Table 6 presents the demographic characteristics and the difference in anthropometric indices between the cases and their community controls. The median age of the participants was 22 (16–40). Men accounted for 51·4% of the subjects. The weight, height (seated and standing) and leg length were significantly smaller in the cases compared with the controls, whereas no difference was seen in thoracic length between the two groups. The average MUAC was lower

**Table 5. Analysis of longer-term post-treatment survival of the hospitalised children traced, and Cox regression exploring the main predictors of death.**

| | N | Died % | HR* (CI 95%) | p value |
|---|---|---|---|---|
| **Nutritional status (WHO standard)** | | | | |
| No Acute Malnutrition** | 133 | 9·0 | Ref | |
| MAM only | 16 | 18·7 | 2·75 (0·76–9·97) | 0·12 |
| SAM only | 105 | 13·3 | 1·63 (0·74–3·57) | 0·22 |
| Stunting and MAM | 74 | 18·9 | 2·35 (1·08–5·11) | 0·030 |
| Stunting and SAM | 1007 | 15·7 | 1·83 (1·01–3·31) | 0·043 |
| **Gender** | | | | |
| Female | 574 | 12·5 | Ref | |
| Male | 761 | 17·0 | 1·44 (1·07–1·92) | 0·013 |
| **Vaccination schedule** | | | | |
| Up to date | 356 | 14·0 | Ref | |
| Not up to date | 979 | 15·4 | 1·13 (0·82–1·57) | 0·96 |
| **Age range (Months)** | | | | |
| < 6 | 31 | 6·5 | Ref | |
| 6–59 | 939 | 15·4 | 2·54 (0·59–10·94) | 0·20 |
| > 59 | 365 | 14·8 | 2·46 (0·58–10·36) | 0·21 |
| **TOTAL** | 1335 | 201(15·1) | | |

HR* = Hazard Ratio adjusted for Nutritional status, Gender, Vaccination schedule and Age range. **CI** = Confidence Interval

**No Acute Malnutrition = No Malnutrition + Stunting only

**Table 6. Differences in growth and body composition between cases and controls.**

| | All participants (931) | Cases (524) | | Controls (407) | | Difference cases-controls (95% CI) | p value |
|---|---|---|---|---|---|---|---|
| | % | % | Mean (SD) | % | Mean (SD) | | |
| **Age (years) Median (Min-Max)** | | | 22 (16–40) | | 22 (16–40) | | |
| **Male** | 51·4 | 52·7 | | 49·8 | | | |
| **Weight (Kg)** | | | 53·5 (7·9) | | 55·1 (7·2) | -1·7 (-2·6 to -0·6) | 0·001 |
| **Height (cm)** | | | | | | | |
| Sitting | | | 112·6 (7·3) | | 113·9 (6·9) | -1·3 (-2·2 to -0·3) | 0·006 |
| Standing | | | 155·9 (8·9) | | 157·6 (8·9) | -1·7 (-2·9 to -0·6) | 0·003 |
| Leg length | | | 91·6 (7·3) | | 93·2 (8·6) | -1·6 (-2·6 to 0·5) | 0·002 |
| Thoracic length | | | 44·6 (7·9) | | 44·9 (9·3) | -0·3 (-1·3 to 0·8) | 0·64 |
| **BMI (Kg/m$^2$)** | | | 22·1 (2·9) | | 22·2 (2·5) | -0·2 (-0·5 to 0·2) | 0.29 |
| < 18·5 | 5·9 | 7·5 | | 3·8 | | | |
| 18·5–24·9 | 81·0 | 79·2 | | 83·2 | | | 0·11 |
| 25–29·9 | 12·2 | 12·3 | | 12·2 | | | |
| ≥ 30 | 0·9 | 1·0 | | 0·8 | | | |
| **MUAC (mm)** | | | 253·5(25·6) | | 256·6 (22·7) | -3·2 (-6·3 to 0·0) | 0·051 |
| **Circumferences (cm)** | | | | | | | |
| Head | | | 54·9 (1·9) | | 55·1 (1·8) | -0·2 (-0·4 to 0·0) | 0·07 |
| Thoracic | | | 83·5 (7·8) | | 83·9 (6·4) | -0·4 (-3·3 to 3·7) | 0·29 |

in the cases than the controls, but of only borderline statistical significance. We did not note any statistically significant difference in head or thoracic circumference between the two groups. The proportion of underweight people tended to be higher among the cases in relation to the controls, whereas that of overweight people and those with an average BMI were similar in the two groups.

## Discussion

The purpose of our study was to find patients 11 to 30 years after their discharge from Lwiro hospital (in eastern DRC) where they were admitted for SAM between 1988 and 2007, and to describe their longer-term survival and their growth to adulthood.

Out of the 1,981 cases entered in the study, 600 were seen by the CHWs and 201 were deceased. We observed that the majority of deaths in former malnutrition patients occur when they are still young (10 or under) and in the 5 years following discharge from hospital. These deaths are often secondary to infectious diseases. Lastly, SAM has lasting negative effects on long-term growth.

To our knowledge, our study is the first in a LIC to have recreated a very large cohort of cases with a history of SAM and to have followed them up a long time (between 11 and 30 years) after their discharge from hospital in a context of endemic malnutrition. Our study was original because it contained a large number of cases with a history of kwashiorkor (70.8%), considered cases who continued to live in an unfavourable environment throughout their life, and examined several anthropometric measurements in order to assess the different aspects of growth. Furthermore, our study opens up avenues for research into the long-term effects of SAM according to these sub-types.

As regards mortality, we noted that the majority of cases died in the first 5 years after discharge from hospital and before their tenth birthday. Moreover, 13·9% of the deaths were associated with a relapse of kwashiorkor and the majority of deaths were secondary to infectious diseases, which are more severe when there is a susceptibility to AM. Our results are consistent

with other studies that have shown that former malnutrition patients remain susceptible to relapses [14–16,42] and various morbidities, especially infectious morbidities [14,42,43]. This medium-term morbidity and mortality seen in our population is due to the fact that after nutritional rehabilitation, former malnutrition patients returned to live in the same unfavourable conditions (unsanitary environment with limited access to drinking water, primary healthcare and a balanced diet) aggravated by the various conflicts in the region. This continued precarious situation therefore created a vicious cycle resulting in a higher risk of morbidity and mortality in those who were already weakened.

It is therefore crucial to ensure proper follow-up after discharge from hospital to ensure the success of the SAM treatment over the medium and long terms. This follow-up should ideally combine a hospital approach with investments in responsive nutrition programmes, such as breastfeeding support, supplemental feeding, and improved farming productivity to minimise food insecurity [42,44]. Moreover, the high prevalence of infectious diseases shows the need to improve the living conditions of households and more especially to improve access to drinking water and sanitation, rapid and effective access to primary healthcare facilities and treatment for acute diseases (such as pneumonia, diarrhoea, measles and malaria), and access to effective immunisation via vaccination. This is still a major challenge in a country with limited resources.

We observed an increased risk of mortality in cases who had AM in relation to cases without AM. This risk became significantly higher when AM (SAM or MAM) was combined with CM. Our results corroborate those of other studies carried out among children in several LMIC [45,46]. In fact, acute malnutrition is the cause of half of all deaths in children aged under 5 in LMIC [1]. Unlike CM, which develops insidiously, AM comes on suddenly and rapidly leads to immune deficiency [47]. As such, children with AM are highly susceptible to life-threatening severe acute infections. The risk of death is even greater if AM is combined with CM [45].

In addition, the risk of death was significantly higher in men than in women. This is consistent with the results of a meta-analysis conducted in LMIC [48]. The greater risk of death in men could be due to natural selection, supported by the theory of evolution modelled by Trivers and Willard and then modified by Wells [49]. According to this theory, natural selection favours a sex ratio of 1·0 to maintain balance. Yet, there are more males conceived than women. Given that this ratio deviates from 1, selection may favour the rarer sex. Consequently, natural selection has favoured men's vulnerability to environmental stress in early life. This vulnerability is most evident in the difficult conditions resulting from premature birth, but can also be seen in infants born at term. Malnutrition, which interacts with infection after birth, is suggested as the fundamental trigger. Consequently, regardless of the improvements in medical care, any environmental stress will always have a more profound effect on men than women in early life, and this will be reflected in a persistent, higher morbidity and mortality in childhood [49].

In terms of growth to adulthood, compared with the community controls, former malnutrition cases had statistically significant low weights, short heights (sitting and standing) and short legs. This would suggest insufficient recuperation, with persistent long-term effects of SAM on growth to adulthood. However, despite this insufficient nutritional recuperation, the thoracic length, and head and thoracic circumference were similar to those of the controls. This suggests that growth of the torso and head was preserved to the detriment of that of the limbs, meaning that the cases' smaller secondary height was due to shorter legs. As such, the growth of SAM survivors could be described as economic, sparing the vital organs to the detriment of the limbs [18,50].

Compared with their community controls, the cases had a borderline statistically significant smaller MUAC. As the MUAC correlates with lean body mass [15,51], the hypothesis would

be that adults with a history of SAM would have a lower lean body mass. This in turn requires a large quantity of micro-nutrients, particularly type I, for its synthesis [52]. Consequently, nutrition education programmes aimed at mothers should focus on the role of energy nutrients while promoting local products.

Despite their low weight and short height, former malnutrition patients have an average BMI similar to that of the controls, suggesting potential catch-up growth (weight-for-height) in adulthood. Although in some ways considered beneficial, this probably reflects rapid weight gain combined with low linear growth. Because former malnutrition patients have muscle wasting and organ dysfunction (liver, heart, kidneys, etc.), rapid weight gain could be secondary to increased body mass. This trend, combined with their already lean body mass and short legs, has been linked with obesity and NCDs later in life [29–31,53]. Nevertheless, this potential post-SAM catch-up growth suggests that improved follow-up care and post-SAM interventions could reverse stunting and underweight status, and give survivors a sufficient lean body mass, thereby reducing their risk of NCDs later in life.

Given the lack of information on the potential long-term health consequences of SAM in early life in LIC, this cohort offers a research opportunity to shed light on several grey areas. But given the age of the participants, we will perhaps not yet see many obvious signs of NCDs, but the precursors will probably be visible. These results could serve as a basis for future programmes to minimise the long-term negative consequences of SAM during childhood.

However, despite our results, certain limitations must be mentioned.

First, certain records found at the beginning had to be excluded from the study because they belonged to patients who had left or died during hospitalisation, were transferred to another facility, were admitted for SAM as an adult, or because they were incomplete. Consequently, we do not know how the data on these subjects might have influenced our results.

Second, a large number of children enrolled (32·6%) in our study had not been traced by the end of the identification phase, 11 to 30 years after their discharge from the HPL. The reason for this may lie in the fact that the region experienced several situations of security instability between 1994 and 2002, with massive population displacements fleeing the conflicts. It is therefore difficult for us to have an opinion on the impact of their data on mortality and long-term growth given that we do not have any information on their outcomes. Nevertheless, we estimate that this would not be significantly different to our main findings given that the characteristics on admission do not differ between those lost to follow-up and the subjects who were traced.

Third, we do not have demographic information concerning infancy, including gestational age, birth weight and height, rate of growth in the first two years of life or data on growth between discharge from hospital and the time when our study was conducted. These items of information could be potential confounding factors, because they are linked with both malnutrition and negative long-term effects [54].

Fourth, we only found controls for three quarters of our cases. This forced us to make group comparisons rather than compare each case with their control. Moreover, although not malnourished in the past, the controls have lived in the same unfavourable conditions and it is difficult for us to say if they are all healthy.

Fifth, the causes of death taken from the registers that we selected are in fact only presumed causes of death given that no autopsy was carried out to accurately confirm the precise cause of death. However, we compared the information given by relatives with that in the medical records to reduce bias.

To conclude, SAM during childhood has lasting negative effects on growth to adulthood. Moreover, these adults have characteristics that may expose them to a risk of NCDs later in life. However, as the potential catch-up growth suggests sufficient recuperation with

appropriate post-SAM interventions, this trend could be reversed and the risk minimised, thereby enabling SAM survivors to thrive.

## Acknowledgments

We express gratitude to CEMUBAC, Foundation Von Buren, Professor Hennart, Yves Mwene-Batu, and all community health workers for the data collection and administrative support during this work. We appreciate the support of Sud-Kivu Health authorities, CRSN authorities, and all village leaders involved in this study.

## Author Contributions

**Conceptualization:** Pacifique Mwene-Batu, Ghislain Bisimwa, Jean Macq, Philippe Donnen.

**Data curation:** Pacifique Mwene-Batu, Gaylord Ngaboyeka.

**Formal analysis:** Pacifique Mwene-Batu.

**Funding acquisition:** Ghislain Bisimwa, Jean Macq.

**Investigation:** Pacifique Mwene-Batu.

**Methodology:** Pacifique Mwene-Batu, Michelle Dramaix, Philippe Donnen.

**Supervision:** Philippe Donnen.

**Validation:** Michelle Dramaix, Daniel Lemogoum, Philippe Donnen.

**Writing – original draft:** Pacifique Mwene-Batu.

**Writing – review & editing:** Ghislain Bisimwa, Gaylord Ngaboyeka, Michelle Dramaix, Daniel Lemogoum, Philippe Donnen.

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
