## [Decision Letter · Decision Letter 0]

20 Jan 2020

PONE-D-19-31748

Follow-up of a historic cohort of children treated for severe acute malnutrition between 1988 and 2007 in Eastern Democratic Republic of Congo

PLOS ONE

Dear Dr. Pacifique Lyabayungu Mwene-Batu,

Thank you for submitting your manuscript to PLOS ONE. After careful consideration, we feel that it has merit but does not fully meet PLOS ONE’s publication criteria as it currently stands. Therefore, we invite you to submit a revised version of the manuscript that addresses the points raised during the review process.

We would appreciate receiving your revised manuscript within 60 days. To enhance the reproducibility of your results, we recommend that if applicable you deposit your laboratory protocols in protocols.io, where a protocol can be assigned its own identifier (DOI) such that it can be cited independently in the future. For instructions see: http://journals.plos.org/plosone/s/submission-guidelines#loc-laboratory-protocols

We look forward to receiving your revised manuscript.

Kind regards,

Gianfranco D. Alpini

Academic Editor

PLOS ONE

Journal Requirements:

2. Please provide additional details regarding participant consent. In the ethics statement in the Methods and online submission information, please ensure that you have specified (a) whether consent was informed and (b) what type you obtained (for instance, written or verbal). As your study included minors, state whether you obtained consent from parents or guardians.

"This study is part of a Research for Development Project entitled: ‘Implementation study of a model of psycho-medico-social care at the health centre level: the case of people with chronic diseases and the mother-child malnourished couple, South Kivu, Democratic Republic of Congo’ and funded by Belgian Development Cooperation through the Académie de Recherche et d’Enseignement Supérieur (ARES)."

Reviewers' comments:

Reviewer's Responses to Questions

**Comments to the Author**

1. Is the manuscript technically sound, and do the data support the conclusions?

Reviewer #1: Yes

Reviewer #2: Yes

2. Has the statistical analysis been performed appropriately and rigorously? 

Reviewer #1: Yes

Reviewer #2: Yes

3. Have the authors made all data underlying the findings in their manuscript fully available?

Reviewer #1: Yes

Reviewer #2: Yes

4. Is the manuscript presented in an intelligible fashion and written in standard English?

Reviewer #1: Yes

Reviewer #2: Yes

5. Review Comments to the Author

Reviewer #1: The paper of Pacifique Mwene-Batu and coauthors explores the long-term outcome of a large cohort of patients affected by severe malnutrition (SAM) in early childhood. They show as this has lasting negative effects on growth to adulthood. Moreover, these adults have characteristics that may expose them to a risk of NCDs later life.

Their effort has to be commended for the number of cases identified and traced until adult age and for the relevant topic investigated.

There are no major concerns from this reviewers perspective.

Minor comments

Are there data on readmissions of patients for SAM over the years other than the incident admissions?

Is it known and can be analysed the impact of medical campaigns of SAM? And data of social context where each individual was going back to after discharge?

It is suggested to structure the results section into paragraphs.

Reviewer #2: This is a very important contribute, with solid statistical analysis, which highlight the long term effect of acute malnutrition. The paper is very well written and discussed.

I think this paper should be highlighted and commented.

Minor comment:

It will be interesting to note if possible the presence of alcohol abuse and associated psychiatric disorders.

6. PLOS authors have the option to publish the peer review history of their article (what does this mean?). If published, this will include your full peer review and any attached files.

Reviewer #1: No

Reviewer #2: No

---

## [Author Response · Author response to Decision Letter 0]

10 Feb 2020

A. Journal Requirements:

I would like to update my funding statement.

Currently, my Funding Statement reads as follows: "The funders had no role in study design, data collection and analysis, decision to publish, or preparation of the manuscript." Instead of that, please put "This study is part of a Research for the Development Project entitled: ‘Implementation study of a psycho-medico-social care model at the health centre level: the case of people with chronic diseases and the mother-child malnourished couple, South Kivu, Democratic Republic of Congo’ and funded by Belgian Development Cooperation through the Académie de Recherche et d’Enseignement Supérieur (ARES). The funder of the study had no role in study design, data collection, data analysis, data interpretation, or writing of the report. "

B. Reviewers' comments:

1.Are there data on readmissions of patients for SAM over the years other than the incident admissions?

R/ At this time, we do not have data on readmissions of patients identified for SAM during their childhood. Given the objective and scope of our study and due to financial and logistical constraints we could not check in all the health facilities in the region if these children were readmitted for SAM. However, among the causes of death in our possession, we observed that around 14% of deaths were secondary to a relapse of SAM. But, the long-term effects of SAM would likely also be influenced by the numbers of hospitalizations. In our future studies, by studying the effects in relation to the subgroups, we believe that this information will be of use to justify certain differences probably observed.

2.Is it known and can be analysed the impact of medical campaigns of SAM? And data of social context where each individual was going back to after discharge?

R/ 

-It is very difficult for us to know and especially to analyze the possible impact of medical campaigns on SAM over such a long period retrospectively due to data unavailable

-In terms of social context, the vast majority of this population lives in a very difficult socio-economic situation, as mentioned in the introduction. We do not have individual data from the specific context in which the children lived after nutritional rehabilitation. Nevertheless, the fact that we have observed long-term persistent effects of SAM during childhood on growth in adulthood is a sufficient proof that they continued to live in the same unfavorable conditions even after nutritional rehabilitation. Furthermore, given their age at admission and the fact that most of them live outside the parental home at the moment, it is difficult for us to say exactly the social context in which they lived after nutritional rehabilitation. However, we will describe in a future study their current socioeconomic context. 

3.It is suggested to structure the results section into paragraphs.

R/ Thank you for the remark. It has been taken into account and incorporated in the text.

4.It will be interesting to note if possible the presence of alcohol abuse and associated psychiatric disorders.

R/ We have data on alcohol consumption in our population study but not on alcohol abuse. However, based on the objective and scope of our work, we opted to use this information for the next studies in which we will examine the risk of cardio-metabolic pathologies in depth. Concerning psychiatric disorders, we have no information now. However, we are conducting a study in which we are measuring the cognitive development of our subjects in adulthood which will allow us to integrate certain tests including the Mini Mental Statu Examination (MMSE) in order to detect cognitive disorders and hence request information related to the psychiatric disorders.

---

## [Decision Letter · Decision Letter 1]

12 Feb 2020

Follow-up of a historic cohort of children treated for severe acute malnutrition between 1988 and 2007 in Eastern Democratic Republic of Congo

PONE-D-19-31748R1

Dear Dr. Pacifique Lyabayungu Mwene-Batu,

We are pleased to inform you that your manuscript has been judged scientifically suitable for publication and will be formally accepted for publication once it complies with all outstanding technical requirements.

With kind regards,

Gianfranco D. Alpini

Academic Editor

PLOS ONE

Additional Editor Comments (optional):

Reviewers' comments:

Reviewer's Responses to Questions

**Comments to the Author**

1. If the authors have adequately addressed your comments raised in a previous round of review and you feel that this manuscript is now acceptable for publication, you may indicate that here to bypass the “Comments to the Author” section, enter your conflict of interest statement in the “Confidential to Editor” section, and submit your "Accept" recommendation.

Reviewer #1: All comments have been addressed

2. Is the manuscript technically sound, and do the data support the conclusions?

Reviewer #1: Yes

3. Has the statistical analysis been performed appropriately and rigorously? 

Reviewer #1: Yes

4. Have the authors made all data underlying the findings in their manuscript fully available?

Reviewer #1: Yes

5. Is the manuscript presented in an intelligible fashion and written in standard English?

Reviewer #1: Yes

6. Review Comments to the Author

Reviewer #1: Authors replied thoroughly to the concerns raised. There are no additional comments from this reviewer.

7. PLOS authors have the option to publish the peer review history of their article (what does this mean?). If published, this will include your full peer review and any attached files.

Reviewer #1: Yes: Marco Carbone

---

## [Editor Report · Acceptance letter]

14 Feb 2020

PONE-D-19-31748R1 

Follow-up of a historic cohort of children treated for severe acute malnutrition between 1988 and 2007 in Eastern Democratic Republic of Congo 

Dear Dr. Mwene-Batu:

I am pleased to inform you that your manuscript has been deemed suitable for publication in PLOS ONE. Congratulations! Your manuscript is now with our production department. 

With kind regards,

on behalf of

Dr. Gianfranco D. Alpini 

Academic Editor

PLOS ONE